computational chemistry

Novichok, nerve agent, toxicity, quantum calculation, acetylcholinesterase

**Author for correspondence:**
Keunhong Jeong
e-mail: doas1mind@kma.ac.kr;
doas1mind@berkeley.edu

This article has been edited by the Royal Society of Chemistry, including the commissioning, peer review process and editorial aspects up to the point of acceptance.

# Theoretical study on the toxicity of 'Novichok' agent candidates

Keunhong Jeong[1] and Junwon Choi[2,3]

[1]Department of Chemistry & Nuclear and WMD Protection Research Center, Korea Military Academy, Seoul 01805, South Korea
[2]Chemical Kinomics Research Center, Korea Institute of Science and Technology (KIST), 5 Hwarangro 14-gil, Seongbuk-gu, Seoul 02792, Republic of Korea
[3]Division of Bio-Medical Science & Technology, KIST School, University of Science and Technology (UST), Seoul 02792, Republic of Korea

KJ, 0000-0003-1485-7235

The identities and properties of 'Novichok' (Russian for 'newcomer' or 'novice') agents allegedly used in the recent terrorist attack in the UK have not been well documented. Although several people previously involved in the synthesis of these materials claimed Novichok agents to be five to eight times more potent than VX, a deadly nerve agent, no open data on these species are currently available. To bridge this gap, we herein performed a theoretical study on several Novichok agent candidates and conducted natural population charge analysis to evaluate the possible mechanisms of their toxicity, suggesting that these agents might promote the ageing and deformation of acetylcholinesterase. Moreover, the reaction of Novichok agents with acetylcholinesterase serine was calculated to be most thermodynamically favoured for Novichok candidate A234. Thus, this work is believed to provide a basis for finding Novichok antidotes and should inspire further detoxification studies to prepare for possible terrorist attacks in the future.

## 1. Introduction

'Novichok' refers to a novel series of nerve agents developed in the Soviet Union and Russia in 1971–1993 [1,2]. Even though the use of these agents on the battlefield or in terrorist attacks has not been proven, one of them is thought to have been recently used for poisoning several people in the UK [3]. Additionally, accidental exposure of Andrei Zheleznyakov (one of the scientists involved in the Novichok development project) to a Novichok agent induced deadly nerve damage that resulted in permanent loss of arm functionality, liver cirrhosis, epilepsy, depression and the inability to read or concentrate [4,5]. Therefore, the tremendous toxicity of Novichok was observed and proven at least in one instance.

The structures of Novichok agents, believed to be developed under the secret 'FOLIANT' programme [6,7], have not yet been

fully disclosed. The information provided by two chemists having experience in this field, namely Mirzayanov and Fyodorov, allowed one to propose some structures for Novichok agents, which were deemed to be five to eight times more toxic than VX [4,6–10].

Novichok agents are thought to feature an organophosphorus core with P=O, P=S or P=Se bonds, featuring fluorine as a leaving group and containing other organic groups such as phosgene oxime and analogues for enhanced toxicity [11–16].

Among some recently proposed organophosphate structures of Novichok agents [16,17], in this study, we focused only on the chemical structures of A230, A232 and A234 proposed by Hoenig [18] and Ellison [19]. The ageing process of these compounds after the inhibition of acetylcholinesterase, which is known to be the loss of one of the groups bound to the phosphorus atom, seems to be much faster than that of the Mirzayanov structure, and secondary reaction of A230, A232 and A234 is possible, which we refer to as a new type of ageing mechanism (see the electronic supplementary material for the structure of A234 proposed by Mirzayanov).

As mentioned above, no studies on Novichok agents have been conducted before, and their high toxicity has therefore not been scientifically explained well enough. Herein, we use quantum mechanical calculations to investigate three representative uncovered Novichok agent candidates and explain their severe harmfulness, additionally suggesting and analysing other possible reactions of these agents.

## 2. Methods

The chemical structures of the studied chemical warfare agents (CWAs) are shown in figure 1. All calculations were carried out using the Gaussian 16W software package [20]. Each starting CWA structure was constructed by PM6 semi-empirical calculations after performing a conformational search using the scan option. Each geometry was refined by density functional theory calculations at the B3LYP/6-311++G(d,p) level of theory without any symmetry constraints, which was previously used for precisely describing several other CWAs (VX, VG, GA, GB and GD) by Zhang et al. [21]. Solvation model based on density (SMD) quantum mechanical aqueous continuum solvation models were applied to predict the stable structures and free energies of aqueous solvated states [22]. The quadratically convergent self consistent field (SCF) procedure was used, and the symmetry was turned off by external request in order to obtain the reliable structures. Harmonic vibrational frequency calculation on each structure was performed to confirm the global minimum structures. Natural bonding orbitals were calculated for obtaining natural population analysis (NPA) charges [23]. Furthermore, to improve the precision of the calculation, we carried out additional calculations at the B3LYP/6-311++G(2d,2p) and M06-2X/6-311++G(2d,2p) levels of theory, respectively. Additionally, it needs to evaluate the level of theory used in this calculation study due to the lack of reliable theoretical information over the studied structures. Therefore, we assessed the quality of the levels of theory which were implemented. Due to the lack of structural information of nerve agents including bond lengths, we compared the theoretical IR spectra data with experimental data. Because the scaling factor depends on its level of theory, $R^2$-values were compared for assessing the level of theories. Interestingly, B3LYP/6-311++G(d,p), B3LYP/6-311++G(2d,2p) and M06-2X/6-311++G(2d,2p) levels of theory showed good agreement on the frequency calculations with high $R^2$-values (0.9925, 0.9942 and 0.9826, respectively) along with small scaling factor (see the electronic supplementary material for detailed information). Although M06-2X/6-311++G(2d,2p) level of theory showed the least correlation between theory and experiment, it is considered as the more expensive and accurate than any other levels in terms of non-covalent interactions. Furthermore, calculated frequency of P=O bond stretch, which is the most important functional group in this study, showed a good match between calculated result and experimental result for all levels of theory. Based on these results, we concluded that all calculation levels could be suitable to be applied in this study.

## 3. Results and discussion

The only information available on the toxicity of Novichok agents is that they are five to eight times more harmful than VX, which is ascribed to the enhanced electrophilicity of phosphorus in their core. In view of the fact that the calculated Novichok agent structures feature fluoride as the leaving group, several well-known fluorinated CWAs were considered for comparison, with the NPA charges of selected CWAs shown in figure 2.

The catalytic mechanism of acetylcholinesterase is based on the formation of a tetrahedral acyl-enzyme intermediate, and phosphate-enzyme intermediate for the chemical agent's reactions.

**Figure 1.** Structures of some well-known CWAs (*a*) and potential Novichok agents (*b*).

Although extensive structural and energy calculations on the transition state of each reaction pathway would be another way of estimating the kinetic parameters, as indicated in several other reports [24,25], charge on the central phosphorus atom is used for evaluating the reactivity of the nerve agent. Therefore, charge in phosphorus atom in organophosphorus compound would be used for toxicity comparison in the limited conditions.

Figure 2 shows that the positive charges on the phosphorus atom in Novichok agents exceed those obtained for other structures, which suggests that the former agents should be more reactive toward the serine residue located at the catalytic site of acetylcholinesterase and hence exhibit increased toxicities. Even though the charge difference among Novichok agents was not substantial, the positive charge in A230 was apparently the largest at different calculation levels, allowing one to suggest that A230 exhibits the highest kinetic reactivity. This deduction was well supported by the fact that A230 featured the shortest P=O bond distance among Novichok agents (see the electronic supplementary material for detailed information). Another important factor to consider is the charge of the carbon in the Novichok-specific oxime group. As shown in figure 2, the impressively high positive charge on this carbon makes it vulnerable to attack by a range of nucleophiles (water or/and amino acid residues) present around the cavity of acetylcholinesterase, which results in accelerated enzyme ageing and deformation. Therefore, the use of Hagedorn oximes (such as Pralidoxime) as antidotes may not be effective, which is another reason of the dreadful Novichok toxicity. Several possible mechanisms of ageing or enzyme deformation can be proposed, depending on whether water, carboxylates or amines (lysine residues) act as nucleophiles (figure 3).

Specifically, the reaction with water molecules produces amino phosphate and carbon dioxide, whereas in reactions with serine, threonine or tyrosine, Novichok acts as a cross-linker to disrupt the enzyme after linking amino acids inside its active site. Reactions with aspartic and glutamic acids produce amino phosphate and carbon dioxide, as in the case of water, while the reaction with lysine results in both cross-linking and amino phosphate formation. All of these reactions, which represent new types of ageing mechanisms, are believed to be very fast because of the high reactivity of phosgene oxime-like structures in Novichok agents formed after binding to serine in the acetylcholine pocket [26] (figure 3).

Herein, the reactivities of A230, A232 and A234 were evaluated by performing a thermodynamic study on their reaction with serine to compare the corresponding $\Delta G$ values at different calculation levels (equations (1)–(3) in table 1).

Even though the $\Delta G$ is not big enough to draw a conclusion on the reactivity, interestingly, a negative $\Delta G$, indicating a spontaneous and thermodynamically favoured reaction, was observed only for A234 with the B3LYP$_1$ (B3LYP/6-311++G(d,p)) level of theory. However, the uncertainties associated with the corresponding calculations and the close-to-zero (small positive) values observed for A230 and A232 make it hard to draw unambiguous conclusions. For the B3LYP$_2$ (B3LYP/6-311++G(2d,2p)) level of calculations, there are no negative $\Delta G$ values, and all negative values of $\Delta G$ are obtained with the

| nerve agent | NPA charge | | |
| --- | --- | --- | --- |
| | B3LYP$_1$ | B3LYP$_2$ | M06-2X$_2$ |
| GA | 2.362 | 2.332 | 2.379 |
| GB | 2.451 | 2.418 | 2.463 |
| GD | 2.453 | 2.418 | 2.463 |
| VX | 2.010 | 2.017 | 2.063 |
| A230 | 2.557 (0.582) | 2.629 (0.591) | 2.684 (0.609) |
| A232 | 2.552 (0.580) | 2.624 (0.590) | 2.679 (0.609) |
| A234 | 2.554 (0.580 | 2.626 (0.590) | 2.679 (0.609) |

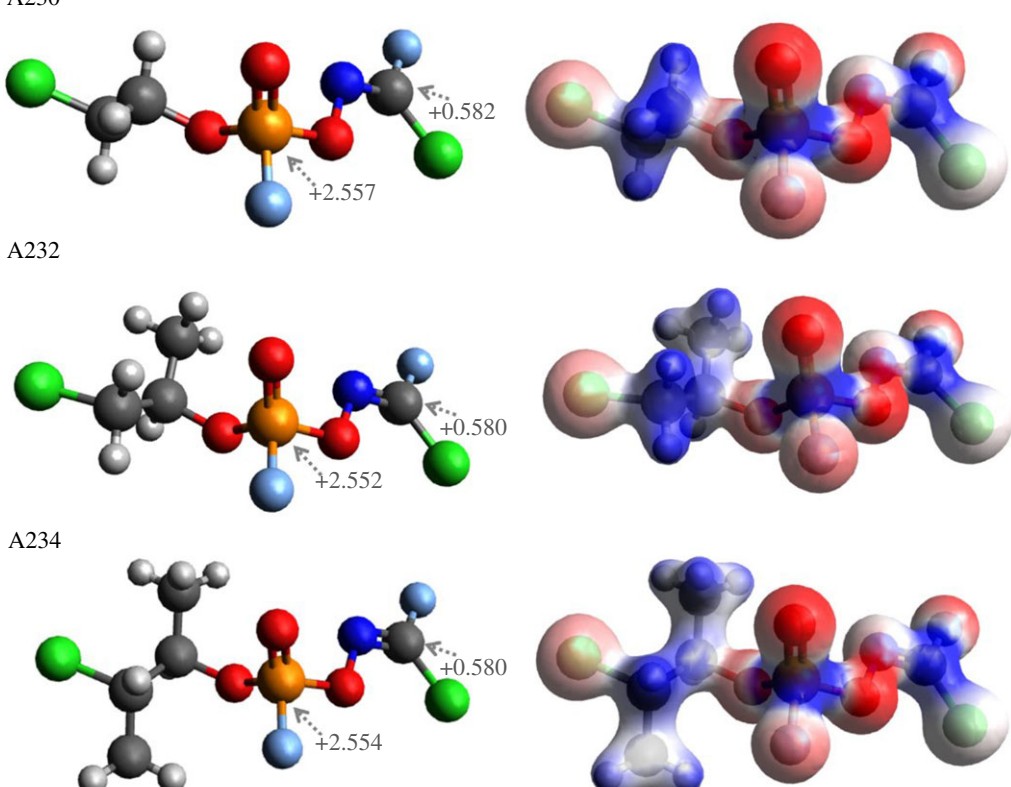

**Figure 2.** Calculated NPA charges on phosphorus and oxime carbons in Novichok agents and those of other CWAs obtained at different calculation levels: B3LYP/6-311++G(d,p) denoted as B3LYP$_1$, B3LYP/6-311++G(2d,2p) denoted as B3LYP$_2$, and M06-2X/6-311++G(2d,2p) denoted as M06-2X$_2$. Electron density image mapped with the electrostatic potential surface.

M06-2X$_2$ (B3LYP/6-311++G(2d,2p)) level of theory. These series of theoretical calculations do not provide its spontaneity in the reactivity; however, A234 has the lowest $\Delta G$ value among all studied structures. One of the reasons for the positive $\Delta G$ values might be calculation errors in the energy of HF, since HF could be stabilized by dissolving in the water. Additional calculations for obtaining the solvation energy of HF were not performed, since this study primarily aimed to compare different Novichok agents. Thus, the obtained results indicate that the reaction with serine is most thermodynamically favoured in the case of A234 among these studied structures. This conclusion is also supported by the obtained P–F bond distance data, since the above reaction has been shown to follow a concerted $S_N2$ mechanism in several theoretical studies (see the electronic supplementary material for detailed information).

# 4. Conclusion

Herein, we used theoretical calculations to determine the relative toxicities of representative Novichok agents and compare them to those of other known CWAs. Although these studies do not provide all

**Figure 3.** Possible mechanisms of the interaction of a Novichok agent with potential electron-rich nucleophiles, (*a*) water, (*b*) hydroxyl groups in amino acid residues, (*c*) carboxylate groups in amino acid residues and (*d*) primary amine groups in amino acid residues in the enzyme after it binds to the serine moiety of acetylcholinesterase.

**Table 1.** Reactions between Novichok agents and serine and the corresponding estimated free energy differences ($\Delta G$). A230 + Serine → A230-Serine   complex + HF   (1)   A232 + Serine → A232-Serine   complex + HF   (2)   A234 + Serine → A234-Serine complex + HF  (3).

| Novichok agent | $\Delta G$ (kcal mol$^{-1}$) | | |
| --- | --- | --- | --- |
| | B3LYP$_1$ | B3LYP$_2$ | M06-2X$_2$ |
| A230 (1) | 0.427 | 2.519 | −0.891 |
| A232 (2) | 0.796 | 1.872 | −1.330 |
| A234 (3) | −0.316 | 1.628 | −1.938 |

toxicity information such as the kinetic data or/and multi-step reaction considerations in biological environment, to the best of knowledge, this study is the first theoretical scientific report on the toxicity of Novichok agents in terms of the enzyme ageing mechanism. Stable CWA structures in aqueous solutions were obtained at a reasonable level of theory after conformational searches. Additionally, the calculated NPA charges and electrostatic potentials revealed the most probable causes of the high toxicity of Novichok agents, and the suggested mechanism implies the occurrence

of additional nucleophilic reactions on the second electrophilic portion of the Novichok agent after its binding to the serine moiety of acetylcholinesterase, which is referred to as the new type of ageing mechanism. Moreover, the data obtained from various calculation levels indicate that A234 is the thermodynamically most reactive nerve agent among the studied structures, whereas the comparison of NPA charges on the phosphorus atom suggests that A230 should exhibit the fastest reaction kinetics. Although theoretical studies only on acetylcholine esterase inhibition do not explain all the toxic effects of the nerve agents, the main plausible reason for their toxicity is the acetylcholinesterase inactivation, thus causing neurotoxicity. Moreover, detoxification methods for developing remedies are focused on the activation of acetylcholine esterase. Significantly, theoretical studies, which can be performed without encountering dangerous situations in dealing with such lethal materials, can yield valuable and reliable data that can be used in practical experimental studies and contribute to the broadening of the scope of the knowledge. Thus, this theoretical study forms a basis for future investigations on the nature of research performed on Novichok agents and their toxicity, and is expected to contribute to the development of suitable detoxification methods with supporting experimental analysis.

Data accessibility. Data analysed in this study are available in the electronic supplementary material.

Authors' contributions. K.J. designed the study and carried out the simulations; J.C. and K.J. studied the possible mechanism and wrote the paper. All authors gave final approval for publication.

Competing interests. We have no competing interests.

Funding. This work was funded by the NRF research fund (NRF-2017R1C1B5016645) and Hwarangdae Research Institute of the Korea Military Academy.

Acknowledgements. The authors acknowledge Dr Yong Han Lee in Agency for Defense Development for valuable discussion.

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
