## [Reviewer comments · Royal Society Open Science]

Review History

RSOS-190414.R0 (Original submission)

Review form: Reviewer 1

Is the manuscript scientifically sound in its present form?

No

Are the interpretations and conclusions justified by the results?

No

Is the language acceptable?

Yes

Is it clear how to access all supporting data?

Yes

Do you have any ethical concerns with this paper?

No

Have you any concerns about statistical analyses in this paper?

No

Recommendation?

Reject

Comments to the Author(s)

This manuscript investigates several possible structures of “Novichok” agents and attempts to evaluate the possible mechanisms of their toxicity. It provides some computational data for these structures in regard to their NPA charges and reaction thermodynamics with serine. However this report suffers from several intrinsic problems (as shown below), and therefore it should be rejected in the current form.

1. For a computational study, reaction kinetics can not be simply inferred based on the positive charge on a certain atom. At least a reaction path, including the energy barrier and transition state(s) should be investigated and provided.
2. 6-311++G** basis set is not sufficient for free energy calculation. Larger basis set should have been used.
3. The Gibbs Free Energy change for A230, A232 and A234 are all very close (Table 1, within 1 kcal/mol), thus it is questionable to conclude that reaction with A234 is the most thermodynamically favored; such a conclusion might easily be proven wrong if larger basis sets or more realistic water models (rather than a continuum) are used.

Review form: Reviewer 2

Is the manuscript scientifically sound in its present form?

Yes

Are the interpretations and conclusions justified by the results?

Yes

Is the language acceptable?

Yes

Is it clear how to access all supporting data?

Yes

Do you have any ethical concerns with this paper?

No

Have you any concerns about statistical analyses in this paper?

No

Recommendation?

Major revision is needed (please make suggestions in comments)

Comments to the Author(s)

Manuscript ID RSOS-190414

Title: Theoretical study on the toxicity of “Novichok” agent candidates

General Comments

The manuscript reported on a theoretical study on the toxicity of “Novichok” agent candidates. The authors have done excellent work and this is the first theoretical study on the subject.

However, a major revision is necessary before the manuscript can be accepted for publication.

Specific comments

The authors must validate the theory used in the computation. Firstly, they should validate the DFT functionals and basis sets used in this study with other functionals and basis set. There should be reproducible results close to the experimental bond length, bond angle and dihedral angles of similar systems.

Review form: Reviewer 3 (Kah-Hui Wong)

Is the manuscript scientifically sound in its present form?

Yes

Are the interpretations and conclusions justified by the results?

Yes

Is the language acceptable?

Yes

Is it clear how to access all supporting data?

Yes

Do you have any ethical concerns with this paper?

No

Have you any concerns about statistical analyses in this paper?

I do not feel qualified to assess the statistics

Recommendation?

Major revision is needed (please make suggestions in comments)

Comments to the Author(s)

Authors must critically analyse the current findings and be able to bridge the gap of their studies and the following reported two studies. Novelty and benefits of current study are the main interests.

(1) Peter R. Chai, Bryan D. Hayes, Timothy B. Erickson & Edward W. Boyer (2018) Novichok agents: a historical, current, and toxicological perspective, *Toxicology Communications*, 2:1, 45-4

(2) Moshiri, M., Darchini-Maragheh, E., & Balali-Mood, M. (2012). Advances in toxicology and medical treatment of chemical warfare nerve agents. *Daru : journal of Faculty of Pharmacy, Tehran University of Medical Sciences*, 20(1), 81. doi:10.1186/2008-2231-20-81

Decision letter (RSOS-190414.R0)

22-May-2019

Dear Dr Jeong:

Title: Theoretical study on the toxicity of "Novichok" agent candidates
Manuscript ID: RSOS-190414

The editor assigned to your manuscript has now received comments from reviewers. We would like you to revise your paper in accordance with the referee and Subject Editor suggestions which can be found below (not including confidential reports to the Editor). Please note this decision does not guarantee eventual acceptance.

Please submit your revised paper before 14-Jun-2019. Please note that the revision deadline will expire at 00.00am on this date. If we do not hear from you within this time then it will be assumed that the paper has been withdrawn. In exceptional circumstances, extensions may be possible if agreed with the Editorial Office in advance. We do not allow multiple rounds of revision so we urge you to make every effort to fully address all of the comments at this stage. If deemed necessary by the Editors, your manuscript will be sent back to one or more of the original reviewers for assessment. If the original reviewers are not available we may invite new reviewers.

Please also include the following statements alongside the other end statements. As we cannot publish your manuscript without these end statements included, if you feel that a given heading is not relevant to your paper, please nevertheless include the heading and explicitly state that it is not relevant to your work.

- Acknowledgements

- Funding statement

Please include a funding section after your main text which lists the source of funding for each author.

RSC Associate Editor:

Comments to the Author:

Three independent reviewers were consulted, and their reports are copied below. On balance, they feel that the work meets the criteria of the journal. Additional experiments are required.

RSC Subject Editor:

Comments to the Author:

(There are no comments.)

Reviewers' Comments to Author:

Reviewer: 1

Comments to the Author(s)

This manuscript investigates several possible structures of "Novichok" agents and attempts to evaluate the possible mechanisms of their toxicity. It provides some computational data for these structures in regard to their NPA charges and reaction thermodynamics with serine. However this report suffers from several intrinsic problems (as shown below), and therefore it should be rejected in the current form.

1. For a computational study, reaction kinetics can not be simply inferred based on the positive charge on a certain atom. At least a reaction path, including the energy barrier and transition state(s) should be investigated and provided.
2. 6-311++G** basis set is not sufficient for free energy calculation. Larger basis set should have been used.
3. The Gibbs Free Energy change for A230, A232 and A234 are all very close (Table 1, within 1 kcal/mol), thus it is questionable to conclude that reaction with A234 is the most thermodynamically favored; such a conclusion might easily be proven wrong if larger basis sets or more realistic water models (rather than a continuum) are used.

Reviewer: 2

Comments to the Author(s)

Manuscript ID RSOS-190414

Title: Theoretical study on the toxicity of "Novichok" agent candidates

General Comments

The manuscript reported on a theoretical study on the toxicity of "Novichok" agent candidates. The authors have done excellent work and this is the first theoretical study on the subject.

However, a major revision is necessary before the manuscript can be accepted for publication.

Specific comments

The authors must validate the theory used in the computation. Firstly, they should validate the DFT functionals and basis sets used in this study with other functionals and basis set. There should be reproducible results close to the experimental bond length, bond angle and dihedral angles of similar systems.

Reviewer: 3

Comments to the Author(s)

Authors must critically analyse the current findings and be able to bridge the gap of their studies and the following reported two studies. Novelty and benefits of current study are the main interests.

(1) Peter R. Chai, Bryan D. Hayes, Timothy B. Erickson & Edward W. Boyer (2018) Novichok agents: a historical, current, and toxicological perspective, *Toxicology Communications*, 2:1, 45-4

(2) Moshiri, M., Darchini-Maragheh, E., & Balali-Mood, M. (2012). Advances in toxicology and medical treatment of chemical warfare nerve agents. *Daru : journal of Faculty of Pharmacy, Tehran University of Medical Sciences*, 20(1), 81. doi:10.1186/2008-2231-20-

Author's Response to Decision Letter for (RSOS-190414.R0)

See Appendix A.

RSOS-190414.R1 (Revision)

Review form: Reviewer 2

Is the manuscript scientifically sound in its present form?

Yes

Are the interpretations and conclusions justified by the results?

Yes

Is the language acceptable?

Yes

Do you have any ethical concerns with this paper?

No

Recommendation?

Major revision is needed (please make suggestions in comments)

Comments to the Author(s)

Journal: Royal Society Open Science

Manuscript ID RSOS-190414.R1

Theoretical study on the toxicity of "Novichok" agent candidates

General comments

The authors have improved the manuscript substantially. However, they have not addressed some major concerns on the theoretical accuracy of the DFT functionals and basis sets used in this work. The discussion is too shallow and not detailed enough. More work needs to be conducted to ensure accuracy of the results and also the theoretical details and the discussion should be expanded. I will recommend major revision to enhance this work.

Specific comments

1. The authors stated: All calculations were carried out using the Gaussian16W software package [19]. In reference [19], in the list of references, the author used Gaussian 09 documentation while in the manuscript they wrote Gaussian 16W. Kindly clarify.
2. The authors must validate the theory used in the computation. Firstly, they should validate the DFT functionals and basis sets used in this study with other functionals and basis set. They should use a similar structural system to compute the bond length, bond angle and dihedral angles and compare the computations to the experimental values for the same structure. They should then select the best functional and basis set after this computation for further theoretical calculation. This will validate the level of DFT theory used.
3. In the theoretical methods there is no mentioned of the accuracy of the computation such as the convergence criteria- Energy, Force and displacements. Also there is no mentioned of whether frequency calculations were conducted in order to determine if the molecules were at a minimum in the potential energy surface. All these procedures must be reported for the benefit of readers.

Decision letter (RSOS-190414.R1)

24-Jun-2019

Dear Dr Jeong:

Title: Theoretical study on the toxicity of "Novichok" agent candidates

Manuscript ID: RSOS-190414.R1

The editor assigned to your paper has now received comments from reviewers. We would like

you to revise your paper in accordance with the referee and Subject Editor suggestions which can be found below (not including confidential reports to the Editor). Please note this decision does not guarantee eventual acceptance.

Please submit a copy of your revised paper before 17-Jul-2019. Please note that the revision deadline will expire at 00.00am on this date. If we do not hear from you within this time then it will be assumed that the paper has been withdrawn. In exceptional circumstances, extensions may be possible if agreed with the Editorial Office in advance. We do not allow multiple rounds of revision so we urge you to make every effort to fully address all of the comments at this stage. If deemed necessary by the Editors, your manuscript will be sent back to one or more of the original reviewers for assessment. If the original reviewers are not available we may invite new reviewers.

RSC Associate Editor:
Comments to the Author:
(There are no comments.)

RSC Subject Editor:
Comments to the Author:
(There are no comments.)

Reviewers' Comments to Author:
Reviewer: 2

Comments to the Author(s)
Journal: Royal Society Open Science
Manuscript ID RSOS-190414.R1
Theoretical study on the toxicity of "Novichok" agent candidates

General comments

The authors have improved the manuscript substantially. However, they have not addressed some major concerns on the theoretical accuracy of the DFT functionals and basis sets used in this work. The discussion is too shallow and not detailed enough. More work needs to be conducted to ensure accuracy of the results and also the theoretical details and the discussion should be expanded. I will recommend major revision to enhance this work.

Specific comments

1. The authors stated: All calculations were carried out using the Gaussian16W software package [19]. In reference [19], in the list of references, the author used Gaussian 09 documentation while in the manuscript they wrote Gaussian 16W. Kindly clarify.
2. The authors must validate the theory used in the computation. Firstly, they should validate the DFT functionals and basis sets used in this study with other functionals and basis set. They should use a similar structural system to compute the bond length, bond angle and dihedral angles and compare the computations to the experimental values for the same structure. They should then select the best functional and basis set after this computation for further theoretical calculation. This will validate the level of DFT theory used.
3. In the theoretical methods there is no mention of the accuracy of the computation such as the convergence criteria- Energy, Force and displacements. Also there is no mention of whether frequency calculations were conducted in order to determine if the molecules were at a minimum in the potential energy surface. All these procedures must be reported for the benefit of readers.

Author's Response to Decision Letter for (RSOS-190414.R1)

See Appendix B.

RSOS-190414.R2 (Revision)

Review form: Reviewer 2

Is the manuscript scientifically sound in its present form?

Yes

Are the interpretations and conclusions justified by the results?

Yes

Is the language acceptable?

Yes

Do you have any ethical concerns with this paper?

No

Recommendation?

Accept as is

Comments to the Author(s)

The authors have improved the manuscript substantially. I recommend acceptance.

Decision letter (RSOS-190414.R2)

16-Jul-2019

Dear Dr Jeong:

Title: Theoretical study on the toxicity of "Novichok" agent candidates

Manuscript ID: RSOS-190414.R2

It is a pleasure to accept your manuscript in its current form for publication in Royal Society Open Science. The chemistry content of Royal Society Open Science is published in collaboration with the Royal Society of Chemistry.

RSC Associate Editor:
Comments to the Author:
(There are no comments.)

RSC Subject Editor:
Comments to the Author:
(There are no comments.)

Reviewer(s)' Comments to Author:
Reviewer: 2

Comments to the Author(s)
The authors have improved the manuscript substantially. I recommend acceptance.

Appendix A

Response to Reviewer's Comments

Theoretical study on the toxicity of “Novichok” agent candidates

Keunhong Jeong, Junwon Choi

Dear Editor, Dr. Laura Smith

I, along with my coauthor, would like to re-submit the attached research article entitled “Theoretical study on the toxicity of “Novichok” agent candidates” for publication in highly-regarded *Royal Society Open Science*. The manuscript ID is RSOS-190414.

The manuscript has been carefully rechecked and appropriate changes have been made in accordance with the reviewers' suggestions after additional extensive quantum calculations. The responses to their comments are attached herewith.

[Reviewer #1]

Comment 1: For a computational study, reaction kinetics can not be simply inferred based on the positive charge on a certain atom. At least a reaction path, including the energy barrier and transition state(s) should be investigated and provided.

Response: We would like to thank the reviewer for this valuable and reasonable comment. We agree with the referee that the reaction kinetic is not simple to derive from the positive charge of phosphorus. And investigating energy barrier and transition states would be the good way of estimating the reaction kinetic. However, we did not consider doing those expensive calculations because charge on the central phosphorus atom is considered as the reactivity descriptor, which can be calculated with reliable method (NBO) with efficient method. Therefore, we added the comments with some citations in the manuscript as following;

Although extensive structural and energy calculations on the transition state of each reaction pathway would be another way of estimating the kinetic parameters, as indicated in several other reports^{23,24}, charge on the central phosphorus atom is used for evaluating the reactivity of the nerve agent.

23 M. L. Mendonca and R. Q. Snurr, *Chem. – A Eur. J.*, 2019, DOI: 10.1002/chem.201900655.

24 J. R. Cox and O. B. Ramsay, *Chem. Rev.*, 1964, 64, 317–352.

Comment 2: 6-311++G** basis set is not sufficient for free energy calculation. Larger basis set should have been used.

Response: We would like to thank the reviewer for this valuable and reasonable comment. We agree with the referee that a larger basis set and appropriate DFT functional would provide more reliable for the estimation of the free energy estimation/prediction. We used B3LYP/6-311++G(d,p) level of theory since several research papers/recent papers/researchers have recently used the same level of theory for the estimation of many reliable properties including the energies. However, we agree with the reviewer's comment; therefore, we performed additionally calculations used with a larger basis set and functional, B3LYP/6-311++G(2d,2p) and M06-2X/6-311++G(2d,2p) level of theory and compared the results to try to provide better insights on the real data.

Comment 3: The Gibbs Free Energy change for A230, A232 and A234 are all very close (Table 1, within 1 kcal/mol), thus it is questionable to conclude that reaction with A234 is the most thermodynamically favored; such a conclusion might easily be proven wrong if larger basis sets or more realistic water models (rather than a continuum) are used.

Response: We are grateful to the reviewer for raising this pertinent point. The energy values are very close; hence, we have additionally described in the discussion. As we noted in the previous response, we performed additional calculations with more expensive methods and interestingly, the +/- sign was found to be different at different levels of the theory., Bbut the thermodynamic favorability is still consistent. Therefore, we have included more discussions on the results of this calculation. In terms of the water model, the first shell water coordination with the continuum model would be a better method as the reviewer pointed out; however, as the reviewer understands, the way the first shell coordinates and the orientations of the coordinating molecules in different structures are all largely related to the energy, which would result in the worst comparison of the data.

For the B3LYP₂ (B3LYP/6-311++G(2d,2p)) level of calculations, there are no negative ΔG values, and all negative values of ΔG are obtained with the M06-2X₂ (B3LYP/6-311++G(2d,2p)) level of theory. These series of theoretical calculations do not provide its spontaneity in the reactivity, however, A234 has the lowest ΔG value among all studied structures.

[Reviewer #2]

Comment : The authors must validate the theory used in the computation. Firstly, they should validate the DFT functionals and basis sets used in this study with other functionals and basis set. There should be reproducible results close to the experimental bond length, bond angle and dihedral angles of similar systems.

Response: We are grateful to the reviewer for raising this pertinent point comment. We used B3LYP/6-311++G(d,p) level of theory since several recent articles have used report the papers used of the same level of theory for estimating many reliable properties including the structures. However, we agree with the reviewer's comment, and therefore, we carried out additionally calculations with a larger basis set and functional, B3LYP/6-311++G(2d,2p) and M06-2X/6-311++G(2d,2p) level of theory and compared the results so as to try to provide better insights on the real data. Thanks to the reviewer's comment, After comparison of the results with those obtained with higher levels of calculations, we could confirmed our conclusions with a higher precision. We appreciate the reviewer's valuable comment.

[Reviewer #3]

Comment : Authors must critically analyse the current findings and be able to bridge the gap of their studies and the following reported two studies. Novelty and benefits of current study are the main interests.

(1) Peter R. Chai, Bryan D. Hayes, Timothy B. Erickson & Edward W. Boyer (2018) Novichok agents: a historical, current, and toxicological perspective, *Toxicology Communications*, 2:1, 45-4

(2) Moshiri, M., Darchini-Maragheh, E., & Balali-Mood, M. (2012). Advances in toxicology and medical treatment of chemical warfare nerve agents. *Daru : journal of Faculty of Pharmacy, Tehran University of Medical Sciences*, 20(1), 81. doi:10.1186/2008-2231-20-81

Response: We are grateful for the reviewer's valuable suggestion. After the careful investigation of these two great papers, we have now included more discussions on the novelty and benefits of our new study with appropriate citations. We emphasize our study by discussing the novelties of the first theoretical study on the reactivity of Novichok candidates and propose new types of aging mechanisms. In terms of benefits, we emphasize the merits of theoretical studies of studying toxic and dangerous materials theoretically before carrying out experiments. Moreover, more expensive calculations with comparison would provide more insights for researchers who develop antidotes to overcome the toxicity of Novichok agents for a peaceful world in the future. We have included several comments in the manuscript, including the following ones:

Among some recently proposed organophosphate structures of Novichok agents^{17,18}, in this study, we focused only on the chemical structures of A230, A232, and A234 proposed by Hoenig and Ellison because their aging, which is known to be the loss of one of the groups bound to the phosphorus atom, after inhibiting acetylcholinesterase seems to be extremely faster than that of the Mirzayanov structure and their secondary reaction is possible, which we refer to as a new type of aging mechanism (see the SI for the structure of A234 proposed by Mirzayanov)

17 M. Moshiri, E. Darchini-Maragheh and M. Balali-Mood, *DARU, J. Pharm. Sci.*, 2012, 20, 81.

18 P. R. Chai, B. D. Hayes, T. B. Erickson and E. W. Boyer, *Toxicol. Commun.*, 2018, 2, 45–48.

Although theoretical studies on only acetylcholine esterase inhibition do not explain all the toxic effects of the nerve agents, the main plausible reason for their toxicity is their rendering the acetylcholinesterase inactive and thus causing neurotoxicity. Moreover, detoxification methods for developing remedies are focused on the activation of acetylcholine esterase. Significantly, theoretical studies, which can be performed without having to encounter dangerous situations in dealing with such lethal materials, can yield valuable and reliable data that can be useful in practical experimental studies and contribute to the broadening of the scope of the knowledge.

We thank you and the reviewers for your thoughtful suggestions and insights, which aided us to improve the manuscript and provide a more balanced and better account of the research. We hope that the revised manuscript is now suitable for publication in your journal.

As before, this manuscript is not under consideration for publication by any other journal or medium, and all authors have agreed to its publication. There are no conflicts of interest to declare.

Thank you for your consideration. I look forward to hearing from you.

Sincerely,

Keunhong Jeong
Department of Chemistry, Korea Military Academy
Seoul 01805, South Korea
Telephone: +82-2-2197-2823
Email: doas1mind@kma.ac.kr

Appendix B

Response to Reviewer's Comments

Theoretical study on the toxicity of “Novichok” agent candidates

Keunhong Jeong, Junwon Choi

Dear Editor, Dr. Laura Smith

I, along with my coauthor, would like to re-submit the attached research article entitled “Theoretical study on the toxicity of “Novichok” agent candidates” for publication in highly-regarded *Royal Society Open Science*. The manuscript ID is RSOS-190414.R1.

The manuscript has been carefully rechecked and appropriate changes have been made in accordance with the reviewers' suggestions after additional extensive quantum calculations. The responses to their comments are attached herewith.

[Reviewer #1]

1. The authors stated: All calculations were carried out using the Gaussian16W software package [19]. In reference [19], in the list of references, the author used Gaussian 09 documentation while in the manuscript they wrote Gaussian 16W. Kindly clarify.

We would like to thank the reviewer for this accurate and valuable comment. We corrected it.

2. The authors must validate the theory used in the computation. Firstly, they should validate the DFT functionals and basis sets used in this study with other functionals and basis set. They should use a similar structural system to compute the bond length, bond angle and dihedral angles and compare the computations to the experimental values for the same structure. They should then select the best functional and basis set after this computation for further theoretical calculation. This will validate the level of DFT theory used.

We would like to thank the reviewer for this valuable and reasonable comment. We agree with the referee that evaluating the level of theory is important in the theoretical study. To our best knowledge, it is impossible to adopt the bond length information of the nerve agent that we investigated here. Therefore, we compared the theoretical IR spectra data with experimental data for GA, GB, GD, and VX. The scaling factor depends on its level of theory; therefore, R-squared values could be used to evaluate the method for choosing reliable level of theory in the theoretical studies. We additionally calculated IR stretches of GA, GB, GD, and VX in gas phase with three levels of theory and compared with reported experimental IR data.

Those analysis showed that the B3LYP/6-311++G(d,p), B3LYP/6-311++G(2d,2p), and M06-2X/6-311++G(2d,2p) level of theory showed good agreement on the frequency calculations with a good R-squared value, and it is also supported by the other publication, which calculated for various nerve agents and its simulants at the B3LYP/6-311++G(d,p) level of theory. Therefore, we added the comments in the manuscript as following with sufficient supporting information in ESM;

Additionally, it needs to evaluate the level of theory used in this calculation study due to the lack of reliable theoretical information over the studied structures. Therefore, we assessed the quality of the levels of theory which were implemented. Due to the lack of structural information of nerve agents including bond lengths, we compared the theoretical IR spectra data with experimental data. Because the scaling factor depends on its level of theory, R-squared values were compared for assessing the level of theories. Interestingly, B3LYP/6-311++G(d,p), B3LYP/6-311++G(2d,2p), and M06-2X/6-311++G(2d,2p) levels of theory showed good agreement on the frequency calculations with high R-squared values (0.9925, 0.9942, and 0.9826, respectively) along with small scaling factor (see ESM for detailed information). Although M06-2X/6-311++G(2d,2p) level of theory showed the least correlation between theory and experiment, it is considered as the more expensive and accurate than any other levels in terms of non-covalent interactions. Furthermore, calculated frequency of P=O bond stretch, which is the most important functional group in this study, showed a good match between calculated result and experimental result for all levels of theory. Based on these results, we concluded that all calculation levels could be suitable to be applied in this study.

3. In the theoretical methods there is no mentioned of the accuracy of the computation such as the convergence criteria- Energy, Force and displacements. Also there is no mentioned of whether frequency calculations were conducted in order to determine if the molecules were at a minimum in the potential energy surface. All these procedures must be reported for the benefit of readers.

We are grateful for the reviewer's valuable suggestion. We added more detailed information in the manuscript on calculation procedures as follows;

The quadratically convergent SCF procedure was used, and the symmetry was turned off by external request in order to obtain the reliable structures. Harmonic vibrational frequency calculation on each structure was performed to confirm the global minimum structures.

We thank you and the reviewer for your thoughtful suggestions and insights, which aided us to improve the manuscript and provide a more balanced and better account of the research. We hope that the revised manuscript is now suitable for publication in your journal.

As before, this manuscript is not under consideration for publication by any other journal or medium, and all authors have agreed to its publication. There are no conflicts of interest to declare.

Thank you for your consideration. I look forward to hearing from you.

Sincerely,

Keunhong Jeong
Department of Chemistry, Korea Military Academy
Seoul 01805, South Korea
Telephone: +82-2-2197-2823
Email: doas1mind@kma.ac.kr